# Removal of a Giant Cyst of the Left Ovary from a Pregnant Woman in the First Trimester by Laparoscopic Surgery under Spinal Anesthesia during the COVID-19 Pandemic

**DOI:** 10.3390/medsci9040070

**Published:** 2021-11-13

**Authors:** Attila Louis Major, Kudrat Jumaniyazov, Shahnoza Yusupova, Ruslan Jabbarov, Olimjon Saidmamatov, Ivanna Mayboroda-Major

**Affiliations:** 1Femina Gynecology Centre, CH-1205 Geneva, Switzerland; 2Department of Obstetrics & Gynecology, University of Fribourg, CH-1700 Fribourg, Switzerland; 3Department of Obstetrics and Gynecology, Urgench Branch of Tashkent Medical Academy, Urgench 220100, Uzbekistan; kudratulla@mail.ru (K.J.); shahnoza.yusupova90@gmail.com (S.Y.); ruslonzabborovl@gmail.com (R.J.); 4Faculty of Tourism and Economics, Urgench State University, Urgench 220100, Uzbekistan; 5Department of Gynecology & Obstetrics, University Hospital of Geneva, CH-1205 Geneva, Switzerland

**Keywords:** laparoscopy, spinal anesthesia, early pregnancy, giant ovarian cyst, COVID-19

## Abstract

This paper reports a case of a 21 year old primigravida at 6 weeks gestation, suffering from important abdominal pain, who was admitted into the medical center with a giant cyst of 28 × 20 cm on her left ovary. A torsion of the ovarian cyst was suspected. Her COVID-19 status was unknown. In view of the emergency of the situation and the COVID-19 pandemic, laparoscopy in spinal anesthesia was performed. The patient remained conscious during the surgical intervention and tolerated it well apart from a slight dyspnea, which was easily eliminated by changing her body position and decreasing the pneumoperitoneum pressure. The ovarian cyst was removed by enlarging the trocar incision. The patient recovered with neither incident nor pregnancy loss. COVID-19-related complications can induce adverse pregnancy outcomes. Under general anesthesia, patients with COVID-19 are at risk of severe pneumonia and of passing their infection to the medical personnel. To avoid such complications in non-specialized centers, laparoscopy should be performed in regional anesthesia. Laparoscopy in spinal anesthesia can be performed safely on pregnant patients by placing them in the proper position, using a low pneumoperitoneum, and monitoring the hemodynamics. During early pregnancy, general anesthesia induces a higher risk of teratogenic effects and of miscarriage.

## 1. Introduction

An immediate surgical intervention is warranted in some emergency situations, such as for patients with appendicitis, necrosis of an appendicular fibroma or, in the example of this case report, torsion of a large ovarian tumor. It is common practice to perform a cesarean section in regional anesthesia, which has become the new standard. Laparoscopy is however still performed commonly in general anesthesia with intubation, as anesthetists and surgeons often lack experience in operating on patients in the Trendelenburg position with a CO_2_-induced pneumoperitoneum. In an earlier paper, the present authors showed encouraging results of laparoscopic surgery performed in spinal anesthesia [1].

Laparoscopy has many advantages over laparotomy and its feasibility was confirmed in surgical pathology [2,3,4,5,6]. The feasibility and safety of performing laparoscopy under general anesthesia during the COVID-19 pandemic was demonstrated recently by comparing its outcomes to those of laparotomy on a group of patients with similar pathologies [2]. Another publication, which has the largest dataset of symptomatic COVID-19 patients, found that compared to general anesthesia, regional anesthesia induces lower pulmonary complication and mortality rates [7]. Performing laparoscopy under regional anesthesia may therefore allow the combination of the advantages of both, which pregnant women may especially benefit from during the COVID-19 pandemic. International guidelines recommend, if possible, performing abdominal surgeries in regional anesthesia [8].

Laparoscopy has many advantages over laparotomy in preventing the transmission of COVID-19 if certain measures are taken. Namely, as a minimally invasive procedure, laparoscopy decreases the risk of viral contamination by blood and of aerosol transmission through the upper airways during the operation, and, thanks to its shorter hospital stay, the post-operative risk of contamination. Performing laparoscopy specifically under regional anesthesia has the added advantage of sparing the need for sedatives, hypnotics, or other intensive care material, which are more prone to shortage during this pandemic [9]. The updated guidelines from the *Society of American Gastroenterology and Endoscopic Surgeons* and the *European Association of Endoscopic Surgery* states that “Although previous research has shown that laparoscopy can lead to aerosolization of blood-borne viruses, there is no evidence to indicate that this effect is seen with COVID-19, nor would it be isolated to minimally invasive surgery (MIS) procedures” [10]. Smoke and aerosol generated during laparoscopy do not increase the risk of COVID-19 transmission compared to open surgery. This point is reinforced by the results of a recent study that examined the presence of coronavirus in surgical smoke generated from tissue incised with an electrocautery scalpel [11]. Viruses could be detected in the surgical smoke but were unable to induce infection in cultured cells. Additionally, the study found that in comparison to no mask use, wearing surgical masks reduced the transmissibility of viral RNA by 99.80%, as detected via PCR.

This paper presents a case report of a pregnant woman in the sixth week of pregnancy undergoing the successful removal of a giant ovarian cyst, which was performed in laparoscopy under spinal anesthesia. The available literature concerning the toxic and the teratogenic effects of anesthesia in early pregnancy is reviewed.

This case report aims to show that this procedure may reduce (1) the teratogenic risk caused by general anesthesia to the embryo in the first trimester of pregnancy; (2) CO_2_-gas-induced risks on the fetus caused by a low pneumoperitoneum pressure; (3) the risk of COVID-19 infection to medical personnel; (4) postoperative complications by avoiding a large incision as is the case in laparotomy

## 2. Materials and Methods

The patient was a 21 year old primigravida with amenorrhea of 6 weeks, who presented at the medical center with lower abdominal pain. Her vital signs were stable with a pulse rate of 72/min and BP of 110–70 mm of Hg. Examination showed her to have mild abdominal distension with lower abdominal tenderness. The uterus was normally sized with marked cervical motion tenderness. Her urine B-HCG test was positive. Her blood test results indicated leukocytes of 8.02 thousand/μL; hemoglobin of 95.3 g/L, hematocrit of 36.33%, platelets of 245 thousand/μL, and an erythrocyte sedimentation rate of 7 mm/h. Urine examination showed an absence of protein, glucose, bacteria, and yeast. Sexually transmitted disease tests were all negative (HIV, Wassermann reaction, hepatitis B, and hepatitis C). The ultrasound examination of the pelvic organs confirmed her to be at 6 weeks of gestation and showing a yolk sac. In the abdominal cavity, a volumetric fluid formation with a diameter of 28 × 20 cm was visualized (Figure 1).

On the basis of the clinical and ultrasound examinations, she was diagnosed with ovarian torsion associated with a giant cyst on the left ovary (Table 1). Surgical emergency was indicated. The proposed surgery and the involved risks for the embryo were explained to the patient and her family, who provided their consent for the intervention.

After examination, the anesthesiologist reported no contraindications with regards to the proposed surgery. Based on the results of the above analyzes and the absence of contraindications, and taking into account the patient’s situation (pregnancy of 6 weeks), the council of obstetricians-gynecologists and anesthesiologists proposed carrying out cyst ovariectomy by performing a laparoscopy under regional (spinal) anesthesia. The surgical procedure was thoroughly explained to the patient and she signed an informed consent form.

Preoperative preparation: for the day preceding the surgery, the patient was recommended to avoid foods that produce intestinal gas and to only eat liquid foods in the evening. A cleansing enema was administered twice: in the evening before the operation and in the morning on the day of the intervention.

Premedication: Atropine 0.1%-1.0, Analgin 50%-2.0, Cerucal 2.0, diphenhydramine 1%-1.0, and Suprastin 1.0 30 min before surgery, IV.

Spinal anesthesia technique: The steps were as follows: the patient was told to lie on her right side. After careful adherence to the rules of asepsis and treating her skin three times with 96° alcohol, local infiltration anesthesia on the area of the proposed puncture was performed with a 5.0 mL 0.5% Novocain solution. A 25-gauge spinal needle was carefully inserted at the L3–L4 level and a hyperbaric solution of lignocaine (heavy) was slowly injected into the subarachnoid space. The patient was immediately transferred to a horizontal position (supine position) and, lowering the head end of the operating table, the patient was transferred to the Trendelenburg position (10–15 degrees) to move the anesthesia in the cranial direction. Eight minutes after the introduction of the lignocaine, pain sensitivity on the skin and the xiphoid process disappeared. The operating table was returned to its original horizontal position. With this regional anesthesia technique, the anesthesia reached approximately the level of Th12-Th11, and above this level, only the analgesic effect remained, and spinal block did not occur.

Hemodynamics were monitored in parallel, indicating the following results: AD-110/70–120/80 mm. hg. art., pulse 80-84-88, respiratory rate 17-18-19 per minute, and SpO2 97–98%. The correction of hemodynamics was carried out with a solution of Mesatone 0.3 mL in 0.9%-100.0 mL isotonic solution. After the fixation of the anesthetic, a Veress needle was introduced into the abdominal cavity to start insufflating CO_2_. Once the intra-abdominal pressure reached 8 mmHg, large and small trocars were inserted. Instruments were introduced using the usual technique. The patient was transferred to the Trendelenburg position at 35–40 degrees. Hemodynamics were strictly controlled (blood pressure, pulse, saturation, respiration). The CO_2_ pressure in the abdominal cavity was left below 8 mmHg. To increase the volume of circulating blood, 500 mL of 0.9% isotonic solution and 500 mL of Ringer’s solution were injected intravenously.

During the revision of the abdominal cavity, a giant cyst was found emanating from the right ovary, limiting the view and access to the small pelvis. The clear serous fluid of the cyst, about 5 liters, was extruded using an aspirator. The uterus showed itself to be spherical and the left appendages appeared normal. A laparoscopic cystovariectomy was performed on the right side. The mass of the cyst was removed from the abdominal cavity through a mini-laparotomic incision where the left trocar was placed.

During the operation, the patient was conscious and answered questions about her state. During the Trendelenburg position, she complained a little about a lack of air, which was promptly resolved by changing her body position and decreasing the CO_2_ insufflation. The hemodynamic parameters were normal. The operation lasted 50 min. During the operation, the patient carefully observed its progress, asked questions, and actively participated in the decision-making. Upon its completion, she was positively surprised. She was thankful that the surgery was done under local anesthesia, allowing her to remain conscious and avoid the adverse effects general anesthesia could have had on the embryo and her.

Postoperative observation and treatment, up to 6 h after the surgery, included the continuous monitoring of respiratory and circulatory functions, control of urine output, parenteral administration of fluids, prevention of thrombosis, and early activation of the patient. There was no need for the appointment of narcotic analgesics or for special stimulation of the intestines. The patient was discharged home in satisfactory condition under the supervision of a local obstetrician-gynecologist 2 days after the laparoscopy.

Pregnancy and Delivery: Immediately after the operation, progesterone was prescribed to prevent miscarriage (200 mg administered 2 times vaginally per day for 2 weeks). The pregnancy continued to develop without complications, with no abnormal findings. The results of ultrasound diagnostics were: Gravidarum-26 weeks, presentation-head. Fetometria: biparietal size—66 mm, abdominal circumference-232 mm, femur length-46 mm, chest circumference-216 mm, fetus weight-930 gr, lateral ventricles of the brain-right-7 mm, left-5 mm, organs are visualized, heartbeat-142 b/min. Gravidarum-31 weeks 1 day, presentation-head. Fetometria: Biparietal size-81 mm, abdominal circumference-265 mm, femur length-57 mm, chest circumference-283 mm, fetus weight-1 kg 164 gr, lateral ventricles of the brain-right-8 mm, left-5 mm, organs are visualized, heartbeat-143 b/min.

The delivery was done by cesarean section on 15 September 2021, as there were signs of fetal distress during an attempted vaginal delivery. A healthy boy of 3720 g was delivered with an Apgar score of 7/8/10 at 38 + 4/7 weeks of pregnancy.

## 3. Discussion

This publication describes the successful removal of a giant adnexal cyst on a patient at 6 weeks gestation by laparoscopy in spinal anesthesia. This paper is the first in the literature to describe the removal of a giant cyst under spinal anesthesia on a patient at her first trimester of pregnancy. No harm to the embryo was observed in the follow-ups during pregnancy.

A video publication from Giampaolino P. et al. in 2019 reported a successful removal of a perforated IUD in the 11th week of pregnancy without specifying the type of regional anesthesia. In the follow-up to this publication, a normal pregnancy in the 23rd week of gestation was reported [5]. In another publication, laparoscopic salpingectomy in spinal anesthesia was performed in the ninth week of gestation because of concomitant intrauterine pregnancy and ruptured ectopic tubal pregnancy [12]. The patient had a cesarean section for fetal distress at 37 weeks of gestation and delivered a healthy female baby of 2800 g with normal postoperative recovery [12].

There are numerous reasons to suspect that the growing fetal brain may be at risk from maternally-administered anesthetic agents. First, most general anesthetic agents are lipophilic and cross the placenta easily. At least in a rodent model, this transplacental transfer was especially related to a directly measurable concentration of isoflurane in the fetal brain [13,14]. Secondly, many non-obstetric surgeries during pregnancy last nearly as long as cesarean deliveries and therefore have a similar impact on the mother and the fetus [15,16,17,18]. Thirdly, excessive concentrations of anesthetic are usually utilized to facilitate uterine quiescence and decrease the threat of preterm labor. Most importantly, the neuro-developmental processes happening during that time—neurogenesis and neuronal migration—are sensitive to environmental and pharmacological influences.

A pregnancy is at high risk of toxicity being exposed to the embryo until the tenth week. During this period most drugs are contraindicated. Recent studies have confirmed accelerated neuronal cell death in immature rodent brains exposed to anesthetic agents, which has raised considerable concern regarding the standard practice of anesthesia [3,4,5]. Many regularly administered anesthetic agents have either N-methyl-D-aspartate (NMDA) receptor blocking or g-aminobutyric acid (GABA) receptor-enhancing properties. NMDA and GABA receptors are commonly delivered throughout the central nervous system and interaction with these receptors is essential for neuronal synaptogenesis, differentiation, and survival during embryonic and fetal development. Neonatal brain development is a complex orchestrated process shaped by the excess production of neurons that subsequently die by apoptosis (a type of programmed cell suicide) as the brain matures. After 28 weeks gestation, the process of neuronal apoptosis is forecasted to include 50% of cortical neurons. During this period of quick brain development, neurons are more susceptible to various metabolic events and, perhaps, anesthetic agents [19]. Studies on rats and mice have clinically proved that nitrous oxide, ketamine, and other NMDA receptor antagonists were the reason for a strengthened apoptosis in immature neurons. Moreover, newborn rats with a 6 h exposure to 0.75% isoflurane [4,5,14], which functions as a GABA receptor stimulant, had widespread neuronal apoptosis and constant memory and learning deficits [15].

Optimal obstetric and neonatal care requires the provision of adequate analgesia for painful procedures. However, anesthetic and analgesic agents have the potential to adversely impact the developing fetal/neonatal brain. In this setting, clinicians must assess the risks and benefits of pharmacologic anesthesia and analgesia for specific indications in this population. However, for the health of the mother and fetus, general anesthesia is required for non-obstetric surgery and cesarean sections in the absence of neuraxial anesthesia.

The advantages of spinal anesthesia are the simplicity of the technique, rapid onset, reduced risk of systemic toxicity, density of anesthetic block, and the postoperative pain relief afforded by neuraxial anesthetic medicine [13].

General anesthesia with intubation induces a higher risk of COVID-19 infection compared to spinal anesthesia [9,20,21] in non-specialized centers [22]. A COVID-19 infection can cause teratogenic toxicity and miscarriage [17,18]. Based on the results of systematic reviews and a meta-analysis of the course of three types of coronavirus infections (SARS, MERS, COVID-19) during pregnancy [19], the following results were obtained. A total of 79 women participated in 19 studies (41 of them with COVID-19) [12,13,20]. Of these women, 39% had a miscarriage and 7% had perinatal death. Research has not yet shown vertical virus transmission across the placenta to be proven [9,10,12,13,21].

Indirect evidence confirms that COVID-19 may cause miscarriage [8,22]. Miscarriages are typically observed with patients having the following COVID-19-like symptoms: fever, inflammation with cytokine storm affecting early implantation and placental function, and a hypercoagulative state during placental infarction [9,23,24]. According to the research [18,19,20], miscarriages were recorded in 38% of the cases (8 of 21). Studies concerning SARS are rare and only 2 out of 19 studies involved miscarriages as an outcome [20]. Data on COVID-19-induced miscarriages are completely missing in the first and second trimester. One paper found that four out of seven pregnant women with SARS during the first trimester of pregnancy had spontaneous miscarriages, and two had elective terminations [19]. The only newborn survivor was delivered at term, and no anomalies were reported. There is only one report of a pregnant woman with MERS, who tested positive at 6 weeks gestation. She was asymptomatic and subsequently delivered a healthy infant at term. There is currently no data on patients infected with COVID-19 during their first trimester of pregnancy and hence the effect of COVID-19 on the fetus in the first trimester is unknown [18,19].

Pelosi et al. presented two cases, in their second trimester of pregnancy, who were operated upon in gasless laparoscopy under epidural anesthesia [25]. Inoue et al. applied locoregional anesthesia combining epidural–spinal anesthesia in the second trimester of pregnancy [26]. They reported three cases of laparoscopically assisted abdominal cystectomy using a gasless method. A Laparolift was used in the first two cases and surgical wire in the third case. All three cases used spinal–epidural anesthesia. Extra-abdominal cystectomy was performed after a mini-laparotomy directly above the ovarian cyst. The case reported by Maiti G.D. et al. shows similarities with this paper’s [12]. Maiti G.D. et al. performed laparoscopic surgery in spinal anesthesia for an extrauterine pregnancy combined with intrauterine pregnancy in the first trimester. In the present case report, laparoscopic surgery was performed under spinal anesthesia in the first trimester of pregnancy, but for a large benign cyst. The peculiarities of this case study are: (1) the removal of a large ovarian cyst during the first trimester of pregnancy; (2) laparoscopic surgery performed under spinal anesthesia; (3) spinal anesthesia in low-pressure pneumoperitoneum with a high degree Trendelenburg position. This approach may be a starting point for larger studies, describing among other things its advantages in the COVID-19 pandemic. Several papers report the benefits of regional anesthesia in laparoscopy and vaginal surgery during the pandemic [27,28,29,30].

Performing surgery on a patient during her first trimester of pregnancy may lead to malformations of the fetus due to the use of narcotic analgesics under intubation or intravenous anesthesia [4,5,31]. The particularity of this paper’s case report lies in the fact that spinal anesthesia was deliberately used to exclude the teratogenic effects of analgesics. Most importantly, a giant cyst (28 cm × 25 cm, with a volume of 5 mL) was removed laparoscopically under a low pressure pneumoperitoneum (below 8 mmHg) in a steep Trendelenburg position(at 35–40 degrees incline).

In the meantime, given the absence of data on COVID-19 infections during early pregnancy, surgeons must err on the side of caution to avoid a possible infection, which may cause miscarriage. Therefore it is judicious to perform laparoscopy in regional anesthesia, especially during the COVID-19 pandemic and/or when specialized equipment is lacking [32]. Such procedures require careful and close coordination between well trained surgeons and anesthetists. This technique is often avoided by unexperienced professionals, because spinal and epidural anesthesia in the Trendelenburg position may affect the thoracal spinal cord and cause respiratory arrest. Prior to this paper, the present authors reported their experience operating on 912 patients with this technique; however, this adverse effect was not encountered in a single patient, even when operated in a steep Trendelenburg position, as the technique was applied properly [32]. Performing a steep Trendelenburg position during a laparoscopy of the pelvic region enables the surgeon to have a better view of the pelvic organs and to perform this surgery under a low pneumoperitoneum pressure. The dyspnea was caused by a high pneumoperitoneum pressure and not by the Trendelenburg position. This technique was well tolerated by patients [32].

By using an intra-abdominal pressure of 8 mmHg, a Trendelenburg position at 35–40 degrees incline, and spinal anesthesia, this technique avoids the potential deleterious effects of carbon dioxide insufflation on the embryo and decreases the possible teratogenic effects of general anesthesia while still preserving the benefits of reduced postoperative pain and a shorter recovery time.

## 4. Conclusions

In three case reports, including this paper’s, no harm to the embryo was observed after laparoscopic surgery under regional anesthesia. Laparoscopic surgery in regional anesthesia is a feasible procedure in the first trimester of pregnancy. It may have advantages during the COVID-19 pandemic not only insofar as it requires fewer drugs and decreases the risk of infection to medical personnel, but also by minimizing the invasiveness of the procedure. Larger studies are needed to confirm these first encouraging observations.

## Figures and Tables

**Figure 1 medsci-09-00070-f001:**
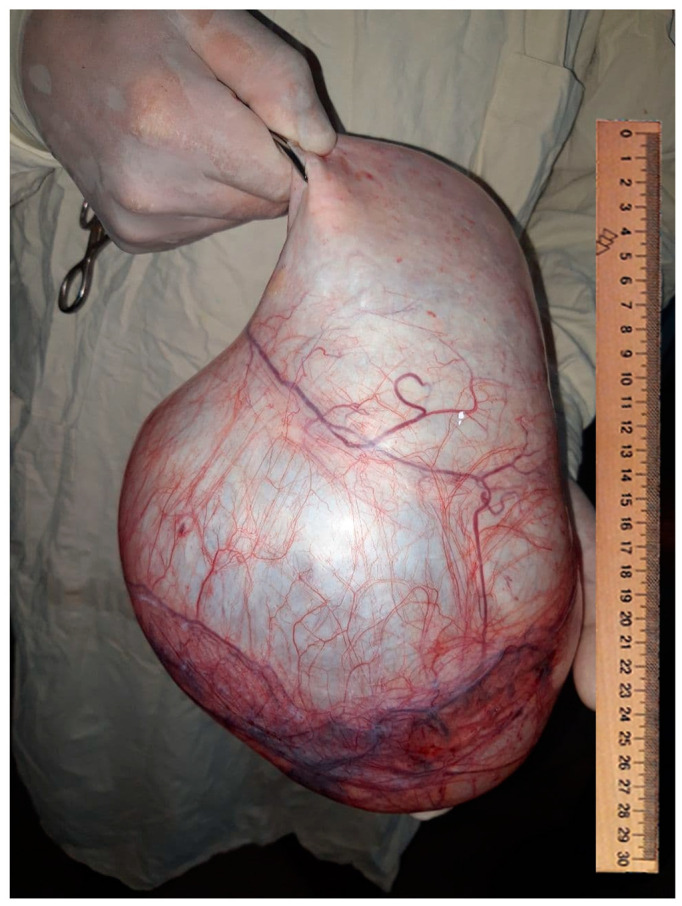
Photo with giant ovarian cyst.

**Table 1 medsci-09-00070-t001:** Results of the ultrasound investigations.

	Parameter	Condition
	Gestational age	6 weeks, 1 intrauterine embryo with positive heart activity was visualized
1	Diameter of gestational sac (GS)	20 mm; corresponds to 6 weeks
2	Embryo with positive heart activity	visualized
3	Crown rump length (CRL) of the embryo	8 mm
4	Heartbeat	+
5	The yolk sac	visualized

Source: self-developed by authors based on clinical observations. +: available.

## Data Availability

Not applicable.

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
