# Peer review of "Removal of a Giant Cyst of the Left Ovary from a Pregnant Woman in the First Trimester by Laparoscopic Surgery under Spinal Anesthesia during the COVID-19 Pandemic"

_medsci, 2021, doi:10.3390/medsci9040070_

Round 1
Reviewer 1 Report
The authors presented a case of a 21-year-old pregnant woman in the 5th week of pregnancy who presented to the emergency department with abdominal pain due to a giant left ovarian cyst of 28x20 cm of the left ovary. Due to suspicion of torsion emergency laparoscopy was performed in spinal anesthesia during COVID-19 pandemic period.
General comment: the case report is very poorly written, besides there is nothing new in this report, all this has already been reported several times in literature. Laparoscopy in spinal anesthesia is well known and established method which was more frequently used during COVID-19 pandemic.
Specific comments:
- Introduction is very short and extremely poorly structured. The authors should provide more information regarding main topic and provide evidences why this topic is important.
- Many unnecessary details in case presentation (e. ‘’during the conversation, the patient received answers to all her questions…. or Table 1 – why the authors mean that all of this laboratory studies are important to the readers, this should be excluded). Process of spinal anesthesia is well known and there is no need of detail description. On the other side, some other important information (e.g. diagnostic tools, radiologic procedures, follow-up…) are not presented.
- Discussion is poorly designed and too general, not focusing on the exact case with many repetitions from existing literature. It should be focused more on the main topic with comparison with similar cases. Repeating of well-known facts from the literature should be avoided.
- Conclusion does not follow from the presented case. These are general statements from literature, not a conclusion of this particular cases.
- References are old and should be updated, similar cases from literature should be cited.
- The main question is how this single case adds to the scientific literature?! There are many similar report, described in the literature that do not differ significantly from this case.
- Quality of English should be improved.
- I never seen that surgeon took a photo with removed specimen and publish it in a scientific Journal. This is absolutely unacceptable. The tumor should be photographed on a flat surface with a scale mark next to it.
Unfortunately, I do not see any benefits for the readers from this case report. This is not enough for publication in international journal.
Author Response
Dear Reviewer,
Authors proivded the responses to your valuable comments, attached please.

Reviewer 2 Report
Well done and interesting paper to read and to know about. Well done the binomious of Laparoscopy and spinal anesthesia
Author Response
Dear Reviewer,
Please, find our responses to your provided comments.

Reviewer 3 Report
Comments on the manuscript:
“Removal of a giant cyst of the left ovary from a pregnant woman by laparoscopic surgery under spinal anesthesia during Covid-19 pandemic: Case report”
This manuscript is a case report describing the removal of a giant ovarian cyst from a 21-year-old woman with primary pregnancy during the 5th week of pregnancy using spinal anesthesia.
The case is well described and I think the article could be published after some improvement. Here are some remarks.
Page 2. Table1 is not called in the text; for hepatitis C: write "Negative" (no “- negative”)
Page 4. Discussion. number the references according to their appearance in the text (change 7, 11, 8, 6).
Lines 138-140: “The recent studies confirm that accelerated neuronal cell death in immature rodent brains exposed to anaesthetic agents has raised considerable concern regarding the standard practice of anaesthesia.” This part is interesting and deserves to have several specific references, not just a review article (8).
Line 178: what is the reference number for Li et al?
Figure 1 is not referred to in the text. Is it really useful to have three photos? Only one seems sufficient to me. Add a scale bar on the chosen photo to appreciate the size of the giant ovarian cyst.
References: check the references and verify if they are presented according to the standards of the journal
Author Response
Dear reviewer,
Please, find our responses to your valuable comments.

Round 2
Reviewer 1 Report
The authors submitted a revised version of the manuscript where they presented a case of laparoscopic tumor removal under general anesthesia. Although the authors performed moderate changes, the overall impression is still poor.
The introduction is still poor, as well as discussion. The authors were advised not to repeat well-known facts, they should discuss their findings and explain why this is important and what makes difference to existing cases from the literature. I do not see significant improvement.
The main question still remains, how this single case adds to the scientific literature?! There are several similar reports, described in the literature which do not differ significantly from this case. Laparoscopic removal of the ovarian tumor under spinal anesthesia is a well known method which has been routinely performed in many centers. I do not see any novelty from this report. An explanation received by the authors that the importance of this case is due to the fact that a big cystadenoma of 25 cm was successfully removed by laparoscopy is premature. Every day in many centers similar tumors are removed by laparoscopy.
Unfortunately, I do not see any benefits for the readers from this case report. This is not enough for publication in international journal.
Author Response
Dear honorable reviewer,
Thanks for your valuable comments. Authors are providing the responses to your comments please, attached below.
If you have any questions, we remain at your disposal.
Best regards,

This manuscript is a resubmission of an earlier submission. The following is a list of the peer review reports and author responses from that submission.